# Acute Ischemic Stroke with Devastating Consequences Can Occur Due to Posterior Sternoclavicular Joint Dislocation

**DOI:** 10.3390/life12111836

**Published:** 2022-11-09

**Authors:** Dana Ghazaleh, Apameh Salari, Leighton Mohl, Madisen Janssen, Kevin Brown, Malik Ghannam

**Affiliations:** 1Department of Neurology, University of Iowa Hospitals and Clinics, 200 Hawkins Dr, Iowa City, IA 52242, USA; 2Department of Neurology, University of Minnesota, 516 Delaware St SE, Minneapolis, MN 55455, USA; 3VA Medical Center, 1 Veterans Drive, Minneapolis, MN 55417, USA

**Keywords:** clavicle fracture, ischemic stroke, secondary stroke

## Abstract

Posterior sternoclavicular joint dislocation is a rare injury requiring prompt diagnosis as it has been associated with the compression of the subclavian and brachiocephalic arteries. We report the case of a 27-year-old male presenting with altered mental status and left hemiparesis found to have life-altering neurologic damage caused by severe orthopedic injury after a fall while intoxicated. Imaging revealed a posteriorly displaced right medial clavicle with surrounding hematoma compressing the brachiocephalic artery causing local thrombus formation and distal embolization, ultimately resulting in malignant cerebral infarction. His hospital course was complicated by cerebral edema requiring decompressive craniectomy, hemorrhagic transformations, brachiocephalic pseudoaneurysm, and the development of remote embolic ischemic infarctions.

## 1. Introduction

Posterior sternoclavicular joint dislocation is a rare injury requiring prompt diagnosis as it has been associated with several complications including the compression of the subclavian and brachiocephalic arteries [1]. This paper outlines a case of malignant cerebral infarction secondary to brachiocephalic artery compression with local thrombus formation and distal embolization, following posterior sternoclavicular joint dislocation.

## 2. Presentation

A 27-year-old male with a past medical history of alcohol use disorder presented to the emergency room after being found encephalopathic. Per his family, he started having new-onset ataxia and dysarthria one day prior to admission. His initial NIHSS was 7 for a decreased level of consciousness, dysarthria, and left hemiparesis. His initial non-contrast head CT showed a large, well-defined infarct of the right MCA territory with cerebral edema causing a 0.7 cm midline shift. A CT angiogram of the head and neck revealed a thrombus in the right brachiocephalic artery at the bifurcation of the right subclavian and common carotid arteries, an occlusion of the right internal carotid artery, and minimal re-opacification of the right MCA branches (Figure 1).

A posteriorly displaced medial right clavicle with surrounding hematoma and compression of the right brachiocephalic artery was also shown, which the patient reports was caused by a fall while intoxicated with alcohol. He was intubated for inability to protect the airway due to worsening mental status and was transferred to our hospital for a higher level of care. Upon arrival, he was evaluated by the hospital’s stroke team and was noted to have worsening mental status, right gaze deviation, and severe left-sided weakness; his NIHSS upon evaluation was 30. Overnight, the patient was noted to have a fixed and nonreactive right pupil which improved after the emergent administration of 23.4% hypertonic saline. Repeat Head CTs demonstrated worsening midline shift and right uncal herniation, and the patient was rushed to the operating room for decompressive craniectomy (Figure 2).

After surgery, the patient was seen by the vascular surgery team who ultimately deferred a thromboendarterectomy of the brachiocephalic artery given the high risk of further embolization, cerebral edema, and hemorrhage. In the following days, he underwent open reduction and internal fixation of the right posterior sternoclavicular joint by orthopedic surgery. His mental status improved to the point that he was extubated and eventually discharged to acute inpatient rehabilitation on warfarin.

The patient soon developed fever and leukocytosis which prompted readmission for infectious workup. Repeat CT angiograms showed a pseudoaneurysm of the right brachiocephalic artery with evidence of active bleeding and associated anterior mediastinal hematoma. Additional findings included a rim-enhancing fluid collection containing a small focus of gas anterior to the sternum as well as the presence of thrombi in the right brachiocephalic artery, subclavian artery, and common carotid artery (Figure 3).

A head CT also revealed a new hemorrhagic infarct in the right occipital lobe. He received a four-factor prothrombin complex concentrate and underwent stent-graft placement of the brachiocephalic artery to achieve hemostasis. After stabilization, he was discharged back to the rehabilitation unit.

Repeat imaging showed evidence of new microhemorrhages, thus delaying the planned continuation of warfarin therapy (Figure 4). Ten days after returning to rehabilitation, the patient developed gaze restriction and was found to have a new medial right thalamus infarction. Repeat CTAs showed persistent thrombi within the proximal right subclavian and right vertebral artery ostium. As the thrombosis was thought to be the source of the embolus, a decision was made to restart anticoagulation therapy. One month after restarting anticoagulation, imaging showed a resolution of the right subclavian artery thrombus. The patient was then transitioned to chronic dual antiplatelet therapy for secondary prevention.

## 3. Discussion

Posterior sternoclavicular dislocation is a rare injury that can present with life-threatening complications given the proximity to great vessels and mediastinal structures [1]. Theoretically, displaced joint and surrounding hematoma can increase the flow velocity and shear stress at the affected vessel wall, leading to intimal damage and subsequent thrombus formation [2,3].

Several cases of the subclavian vein, brachiocephalic vein, and jugular vein injury have been reported as direct complications of this injury [4,5,6]. There have also been several reports of subclavian arterial thrombosis and subsequent cerebral and cerebellar infarcts secondary to cervical rib compression [7,8].

The mechanism of embolization associated with subclavian artery thrombosis is yet to be understood, but in the majority of the literature, it is thought to be secondary to the retrograde propagation of thrombi through the carotid artery as evidenced by this case [8,9]. The larger diameter of the carotid artery compared to the vertebral artery may contribute as well [3,10].

In the present case, the mechanism of injury leading to sternoclavicular dislocation is not certain, although it was reported that he fell from a ladder preceding the injury. The mechanism of cerebral embolization is more clear, as, on the initial imaging, there was evidence of a thrombus in the right brachiocephalic artery. Having thrombosis in the brachiocephalic artery with extension to the common carotid artery leads to embolization and MCA territory ischemic stroke. At the same time, the extension of thrombosis into the subclavian artery put the patient at risk of developing posterior circulation stroke as well. The severity of brain edema requiring decompressive surgery and the development of new intracranial hemorrhage delayed anticoagulation and stent placement, which eventually led to posterior circulation stroke.

## 4. Conclusions

In summary, cerebral infarctions due to a traumatic subclavian arterial thrombosis are unusual; however, when present, the consequences for the brain tissue involved can be serious. It may therefore be wise in clinical situations of trauma involving dislocated clavicles to undertake CT angiography to address the potential vascular damage that might occur.

## Figures and Tables

**Figure 1 life-12-01836-f001:**
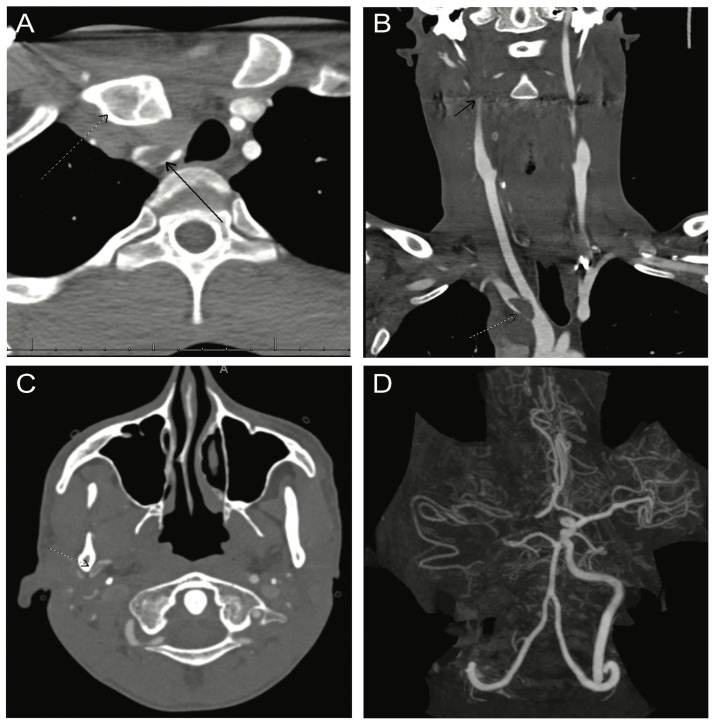
CT angiography demonstrating PSS ((**A**) dotted arrow) with surrounding hematoma causing brachiocephalic artery compression. Thrombus was observed at the site of compression ((**A**) black arrow; (**B**) dotted arrow) with distal emboli within the internal ((**B**) black arrow) and external (**C**) carotid branches, including persistent right MCA occlusion (**D**).

**Figure 2 life-12-01836-f002:**
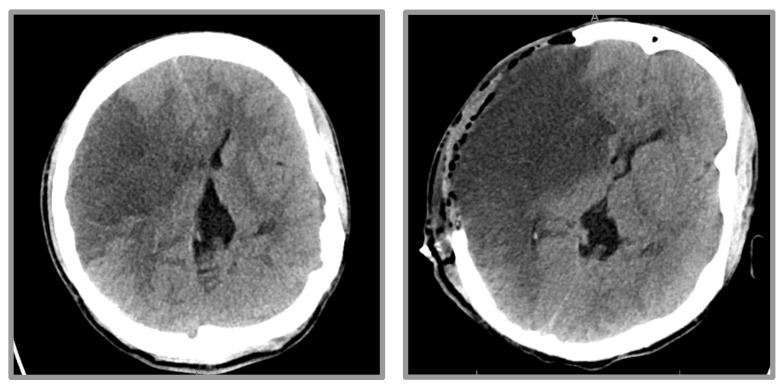
CT head pre- and post-decompressive hemicraniectomy. (**Left Panel**) Evolving large territory subacute infarct affecting much if not all of the right MCA territory. There is a noticeable left shift of midline structures. (**Right Panel**) Interval postsurgical changes of decompressive right craniectomy revealing an edematous right cerebral hemisphere that protrudes beyond the margins of the craniectomy.

**Figure 3 life-12-01836-f003:**
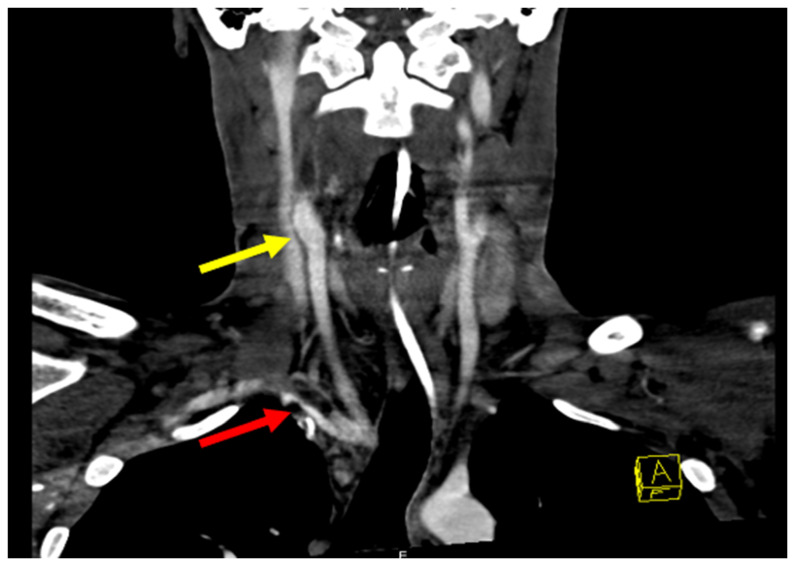
Angiography revealing pseudoaneurysm of right IC and thrombus (yellow arrow). Occluded right internal carotid from the bifurcation at the neck to the skull base and possibly further. Narrowing at the origin of the right vertebral artery from nearly occlusive thrombus in the right subclavian artery. A large pseudoaneurysm of the right brachiocephalic artery is also visible (red arrow).

**Figure 4 life-12-01836-f004:**
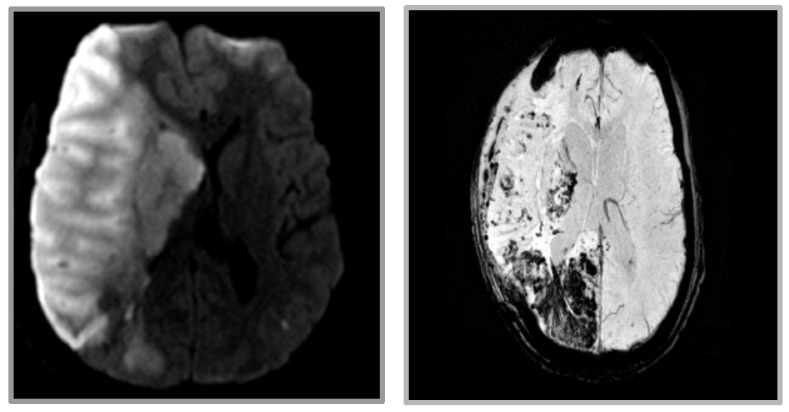
MRI revealing extensive infarct in the right frontal, parietal, occipital, and deep nuclei. (**Left**) DWI imaging. (**Right**) GRE imaging revealing extensive hemorrhagic conversion.

## Data Availability

All the data supporting our findings are contained within the manuscript.

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
