# Peer review of "Acute Ischemic Stroke with Devastating Consequences Can Occur Due to Posterior Sternoclavicular Joint Dislocation"

_life, 2022, doi:10.3390/life12111836_

Round 1
Reviewer 1 Report
the paper is clear and interesting. I would recommend to publish it as it is.
Author Response
Thank you for your comment. Much appreciated.
Reviewer 2 Report
dear authors, this is an interesting case report, however some issues need clarification. You should comment more and argue whether someone should have intervened in the innominate artery at an initial stage. Even the risk of embolism could be prevented if someone had open the area of carotid and put a clamp on it. Or if endovascular approach would be an option with a covered stent graft (kissing stent in the bifurcation)
Additionally, elaborate why Warfarin was prescribed, which is a medication that cannot always be easily controlled. You could change to LMWH or DOAC.
Figure 3 does not clearly show the pseudoaneurysm, thus add an arrow or add a more illustrasive one.
In the discussion area you could add some cases that Angiovac was used to retrieve thrombus, while you sould also comment in endovascular approach of this pathology
Author Response
- You should comment more and argue whether someone should have intervened in the innominate artery at an initial stage. Even the risk of embolism could be prevented if someone had open the area of carotid and put a clamp on it. Or if endovascular approach would be an option with a covered stent graft (kissing stent in the bifurcation):
This is a sub-occlusive thrombus, the best management would be medical treatment with anticoagulation or anti-platelet therapy to prevent further progression in the thrombus size and distal embolization. Endovascular therapy was considered to be the second line in this case, given the high risk associated with endovascular intervention in this case as distal embolization would be not uncommon with this approach.
- Elaborate why Warfarin was prescribed, which is a medication that cannot always be easily controlled. You could change to LMWH or DOAC:
Thank you for the important question. The practice patterns at the particular hospital among the particular neuro-vascular group is to utilize agents which at that point in time had an easy reversible option in case of bleeding issues. Over time, at our center as with many centers, these practice patterns are changing to favor DOACs (importantly there was not a reversal plan for DOACs at the time of this patient's case).
- Figure 3 does not clearly show the pseudoaneurysm, thus add an arrow or add a more illustrasive one:
Thank you, we appreciate that this may not be obvious in the figure as it is presented given there is abundant vascular in view. We are including an edited figure with a red arrow at the right brachiocephalic aneurysm and clot as well as the R. carotid artery pseudoaneurysm in the figure as a yellow arrow. (Please see the attachment below).
- In the discussion area you could add some cases that Angiovac was used to retrieve thrombus, while you should also comment in endovascular approach of this pathology:
The authors reviewed the literature and was unable to relate to the suggested treatment approach for this case.

Reviewer 3 Report
In this case report, the authors presented a case of 27-year-old male patient, presenting with left hemiparesis and altered mental status caused by severe orthopedic injury after a fall while intoxicated.
A malignant right MCA infartion was detected. In etiological work-up, posteriorly displaced right medial clavicle with surrounding hematoma compressing the brachiocephalic artery causing local thrombus formation and distal embolization was found.
The authors noted that major complications developed during the follow-up period. Finally, the patient, who experienced the 2nd stroke and hemorrhagic complications, eventually stabilized and was followed up with dual antiplatelet therapy.
This case report is clinically interesting, rare and well written, visual examples have been instructive and will contribute to the literature.
Author Response
We appreciate your supportive comment.
Reviewer 4 Report
Acceptable for publication. No any major flaw in the manuscript. Except hanging paragraph in page 5 (before discussion)- need to check
Author Response
Thank you for your comment. The "hanging" paragraph will be edited.
Round 2
Reviewer 2 Report
dear authors, this is an interesting case report, however there are some issues that need clarification. Why did not your vascular team intervene? Why did you choose warfarin? There are more medical options with more stable behavior. You should elaborate on literature on endovascular treatment of such procedures.
Author Response
Thank you for the valid and important point. Given the young age of this patient, he was a fast progressor with poor collaterals. The initial CT scan at the time of presentation had an already established infarct with ASPECT score less than 6. Based on the current guidelines in practice, mechanical thrombectomy is not indicated in such situations. We understand this is an ongoing research question regarding doing mechanical thrombectomy in large core strokes. We are waiting for the result of SELECT2 and TESLA clinical trials to hopefully clarify this question. Additionally, in this case in particular, the clot burden was significant and mainly in the ICA. Therefore endovascular treatment will carry more risk.
Regarding warfarin, here is our answer that we shared before. Let us know if you need more clarification: The practice patterns at the particular hospital among the particular neuro-vascular group is to utilize agents which at that point in time had an easy reversible option in case of bleeding issues. Over time, at our center as with many centers, these practice patterns are changing to favor DOACs (importantly there was not a reversal plan for DOACs at the time of this patient's case).